

# Modal Properties and Stability of Bend-Twist Coupled Wind Turbine Blades

Alexander R. Stäblein[1], Morten H. Hansen[1], and David R. Verelst[1]

[1]Technical University of Denmark, Department of Wind Energy, Frederiksborgvej 399, 4000 Roskilde, Denmark.

*Correspondence to:* Alexander R. Stäblein (alexander@staeblein.com)

**Abstract.** Coupling between bending and twist has a significant influence on the aeroelastic response of wind turbine blades. The coupling can arise from the blade geometry (e.g. sweep, prebending or deflection under load) or from the anisotropic properties of the blade material. Bend-twist coupling can be utilised to reduce the fatigue loads of wind turbine blades. In this study the effect of material based coupling on the aeroelastic modal properties and stability limits of the DTU 10 MW Reference Wind Turbine are investigated. The modal properties are determined by means of eigenvalue analysis around a steady-state equilibrium using the aero-servo-elastic tool HAWCStab2 which has been extended by a beam element that allows for fully coupled cross-sectional properties. Bend-twist coupling is introduced in the cross-sectional stiffness matrix by means of coupling coefficients that introduce twist for flapwise (flap-twist coupling) or edgewise (edge-twist coupling) bending. Edge-twist coupling can increase or decrease the damping of the edgewise mode relative to the reference blade, depending on the operational condition of the turbine. Edge-twist to feather coupling for edgewise deflection towards the leading edge reduces the inflow speed at which the blade becomes unstable. Flap-twist to feather coupling for flapwise deflections towards the suction side increase the frequency and reduce damping of the flapwise mode. Flap-twist to stall reduces frequency and increases damping. The reduction of blade root flapwise and tower bottom fore-aft moments due to variations in mean wind speed of a flap-twist to feather blade are confirmed by frequency response functions.

## 1 Introduction

Structural coupling of the flap- or edgewise bending and twist of wind turbine blades has a considerable influence on the aeroelastic response. The coupling creates a feedback loop between the aerodynamic forces, which induce bending in the blade, and the angle of attack, which determines the aerodynamic forces.

Bend-twist coupling can arise from the blade geometry (geometric coupling) or from the anisotropic blade material (material coupling). Geometric coupling is the result of a curved blade geometry (e.g. from prebend, load deflection or sweep) which induces additional torsion when the blade is loaded. Elastic coupling results from the fibre direction in the spar cap and/or skin of the blade. If fibre reinforced plastic laminates are loaded transverse to their principle axes, normal and shear strains become coupled. The coupling transcends to the cross section level where it can result in the coupling of beam bending and twist. Bend-twist coupling can be utilised to tailor the aeroelastic response of wind turbine blades. Early studies on bend-twist coupled blades investigate twisting towards a larger angle of attack for flapwise deflection towards the suction side of the



blade to reduce lift by stalling the aerofoil (flap-twist to stall coupling). With the development towards pitch regulated turbines, twisting towards a smaller angle of attack has also been investigated (flap-twist to feather). The motivation behind bend-twist coupling in wind turbine blade applications has mainly been load alleviation. Fatigue load reductions in the range of 10-20% have been reported for flap-twist to feather coupled blades (Lobitz et al., 1999; Lobitz and Veers, 2003; Verelst and Larsen,
2010; Bottasso et al., 2013).

Apart from the intended load alleviation, bend-twist coupling also affects the aeroelastic modal properties (i.e. frequency, damping, mode shapes) and stability of the blade. Hong and Chopra (1985) investigate the aeroelastic stability of coupled helicopter composite blades using an eigenvalue approach. The structure is modelled by a finite element beam formulation that integrates the strain energies over the cross section, thus explicitly considering the fibre layup. The aerodynamic forces are
assumed quasi-steady. A linearisation of the rotor blade around a steady state equilibrium point is used to obtain the modal properties by means of an eigenvalue analysis. Hong and Chopra report reduced frequencies for edge-twist coupled blades. Twist to feather for edgewise deflection towards the leading edge increases the damping of the edgewise mode. Damping reduces for edge-twist to stall. The authors conclude that edge-twist coupling has an appreciable influence on stability. Twist to feather for flapwise deflection towards the suction side of the aerofoil increases the frequency and reduces the damping of
the flapwise mode. The frequency reduces and damping increases for twist to stall. Lobitz and Veers (1998) investigate the aeroelastic stability of flap-twist to feather and stall coupled blades by casting Theodorsens equations of the aerodynamic lift and moment into pseudo time domain and applying the principle of virtual work to obtain aerodynamic mass, damping and stiffness matrices. The aerodynamic matrices are subsequently combined with the structural matrices to formulate an eigenvalue problem. Lobitz and Veers report a moderately reduced flutter speed for twist to feather coupled blades while divergence
becomes critical for twist to stall. Rasmussen et al. (1999) investigate the damping of a blade section in attached and separated flow. The edge- and flapwise directions of vibration are prescribed and coupled with in phase and counter phase pitch motion. Aerodynamic damping is obtained by integrating the aerodynamic work over one cycle of oscillation. For attached flow, edge-twist to feather coupling reduces the damping for edgewise vibration directions between the inflow and the rotor plane. For edge-twist to stall coupling the damping increases. Flap-twist to feather coupling reduces damping while damping increases
for flap-twist to stall coupling. Lobitz (2004) investigates the flutter speed of an uncoupled and a bend-twist to feather coupled MW-sized wind turbine blade with quasi-steady and unsteady aerodynamic models, applying the Theodorsens approach. Lobitz shows that quasi-steady flutter speeds are significantly lower than the flutter speeds obtained with unsteady aerodynamics. The flutter speed of the coupled blade is moderately lower than of the uncoupled blade. Kallesøe and Hansen (2009) investigate the effect of finite steady state blade deflections on the aeroelastic stability of the NREL 5 MW Reference Wind Turbine (RWT)
(Jonkman et al., 2009) using an eigenvalue approach. A geometric nonlinear finite element beam model and aerodynamic forces obtained from Blade Element Momentum theory are used to find a steady state equilibrium of the turbine. After linearisation of the turbine around the steady state operational point and adoption of a Beddoes-Leishman type dynamic stall model, an eigenvalue analysis is carried out. Rotor dynamics are considered by means of a Coleman transformation. The eigenvalue approach has been implemented in the software tool HAWCStab2 (Hansen, 2004). Kallesøe and Hansen observe a slight
reduction in the flutter limit for deflected blades due to coupling of the edgewise and torsional component. Hansen (2011)



investigates the aeroelastic response of backward swept blades using the eigenvalue approach. Hansen concludes that the backward sweep, which induces flap-twist to feather coupling, mainly influences the flapwise mode and has little influence on edgewise vibrations. Aeroelastic frequencies of the flapwise mode increase while the flapwise damping and the flutter speed reduce with sweep. Hayat et al. (2016) investigate the flutter speed of the NREL 5 MW turbine with flap-twist to feather coupled blades using time domain analysis. Hayat et al. report a slightly reduced flutter speed if coupling is introduced by changing the fibre direction of the glass fibres. If coupling is achieved by using carbon fibres the flutter speed increases due to the higher stiffness of the blade. Stäblein et al. (2016a) investigate the aeroelastic modal properties and stability limits of an edge- and flap-twist coupled blade section using eigenvalue analysis. The authors conclude that damping increases for edge-twist to feather coupling and reduces for twist to stall. Flap-twist to feather increases the frequency and reduces damping while twist to stall has the opposite effect. Stäblein et al. show that edge-twist coupling can result in aeroelastic flutter if the torsional component of the coupled edge-twist mode becomes large enough to enable the formation of an edge-twist flutter mode. Flap-twist to feather leads to a moderate reduction of the classical flutter speed while flap-twist to stall coupling results in divergence.

In this paper the aeroelastic modal properties and stability limits of the DTU 10 MW RWT (Bak et al., 2013) with bend-twist coupled blades are investigated. Coupling is introduced in the cross-section stiffness matrix by means of a coupling coefficient as proposed by Lobitz and Veers (1998). The aeroelastic modal properties and stability limits of both, edge- and flap-twist coupled blades are investigated by means of eigenvalue analysis around a steady-state equilibrium using the aero-servo-elastic tool HAWCStab2. For the analysis with fully coupled cross-section stiffness matrices the beam element of Kim et al. (2013) has been implemented in HAWCStab2.

## 2 Methods

### 2.1 Introduction

The modal properties of the DTU 10 MW RWT are investigated using the aero-servo-elastic code HAWCStab2 (Hansen, 2004). HAWCStab2 calculates the steady state response (including large blade deflections) at an operational point (a combination of wind speed, rotational speed and pitch angle) assuming an isotropic rotor (i.e. no wind shear, yaw, tilt, turbulence, tower shadow or gravity). The aerodynamic forces are based on Blade Element Momentum theory and include tip loss. An analytical linearisation around the steady state is used to determine the modal frequency and damping of the turbine by means of eigenvalue analysis. The linearisation includes the effects of shed vorticity, dynamic stall and dynamic inflow. The periodicity of the system is handled using the Coleman transformation.

To allow for the analysis of anisotropic cross-sectional properties the beam element proposed by Kim et al. (2013) has been implemented into HAWCStab2. The two-noded element assumes polynomial shape functions of arbitrary order where the shape function coefficients are eliminated by minimizing the elastic energy of the beam while satisfying the boundary conditions. The beam element formulation is recapitulated in this section.

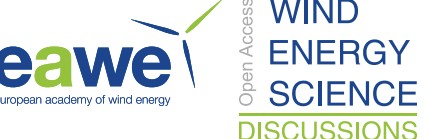

## 2.2 Kinematic Assumptions

The element coordinate system has its origin at the first node of the element. The beam axis $z$ is along the length of the beam, pointing towards the second node. Axes $x$ and $y$ define the cross-sectional plane of the beam. The lateral displacements $u_x, u_y, u_z$, and the rotations $\theta_x, \theta_y, \theta_z$ along the beam axis $z$ are expressed as $N-1$ order polynomials $\sum_{i=0}^{N-1} a_i z^i$. In matrix notation the displacements and rotations along the beam can be expressed as

$$\boldsymbol{u}(z) = \boldsymbol{N}(z)\boldsymbol{\alpha} \tag{1}$$

where $\boldsymbol{u}(z) = \{u_x,\ u_y,\ u_z,\ \theta_x,\ \theta_y,\ \theta_z\}^T$ is the vector of the beam displacements and rotations,

$$\boldsymbol{N}(z) = \left[ \underset{6\times 6}{\boldsymbol{I}}, \ z \underset{6\times 6}{\boldsymbol{I}}, \ z^2 \underset{6\times 6}{\boldsymbol{I}}, \ \dots, \ z^{N-1} \underset{6\times 6}{\boldsymbol{I}} \right] \tag{2}$$

is the polynomial matrix where $\underset{6\times 6}{\boldsymbol{I}}$ are $6\times 6$ identity matrices, and $\boldsymbol{\alpha}$ is the vector of the $6N$ polynomial coefficients which will be called generalised degrees of freedom.

## 2.3 Elastic Energy & Strain Displacement Relation

Assuming plane sections to remain plane a beam strain vector $\boldsymbol{\varepsilon} = \left\{ \frac{\partial u_x}{\partial z} - \theta_y, \ \frac{\partial u_y}{\partial z} + \theta_x, \ \frac{\partial u_z}{\partial z}, \ \frac{\partial \theta_x}{\partial z}, \ \frac{\partial \theta_y}{\partial z}, \ \frac{\partial \theta_z}{\partial z} \right\}^T$ can be introduced. Together with the $6\times 6$ cross-section stiffness matrix $\boldsymbol{K}_{cs}$ the elastic energy $U$ of the beam can be written as

$$U = \frac{1}{2} \int\limits_{z=0}^{L} \boldsymbol{\varepsilon}^T \boldsymbol{K}_{cs} \boldsymbol{\varepsilon} \, dz \tag{3}$$

The beam strain vector can be expressed in terms of the generalized degrees of freedom

$$\boldsymbol{\varepsilon} = \underbrace{\left( \boldsymbol{B}_0 \boldsymbol{N} + \frac{\partial}{\partial z} \boldsymbol{N} \right)}_{\boldsymbol{B}} \boldsymbol{\alpha} \tag{4}$$

where

$$\boldsymbol{B}_0 = \begin{bmatrix} 0 & 0 & 0 & 0 & -1 & 0 \\ 0 & 0 & 0 & 1 & 0 & 0 \\ 0 & 0 & 0 & 0 & 0 & 0 \\ 0 & 0 & 0 & 0 & 0 & 0 \\ 0 & 0 & 0 & 0 & 0 & 0 \\ 0 & 0 & 0 & 0 & 0 & 0 \end{bmatrix} \tag{5}$$

is a transformation and $\boldsymbol{B}$ the strain-displacement matrix. Combining Equations (4) and (3) the elastic energy becomes

$$U = \frac{1}{2}\boldsymbol{\alpha}^T \underbrace{\int\limits_{z=0}^{L} \boldsymbol{B}^T \boldsymbol{K}_{cs} \boldsymbol{B} \, dz}_{\boldsymbol{D}} \ \boldsymbol{\alpha} \tag{6}$$

where $\boldsymbol{D}$ is the beam element stiffness with respect to the generalized degrees of freedom $\boldsymbol{\alpha}$.



## 2.4   Compatibility & Order Reduction

The generalized degrees of freedom are obtained by substituting a part of $\boldsymbol{\alpha}$ denoted $\boldsymbol{\alpha}_1$ by the nodal degrees of freedom $\boldsymbol{d}$ and determine the remainder of $\boldsymbol{\alpha}$ denoted $\boldsymbol{\alpha}_2$ by minimizing the elastic energy. Compatibility with the nodal degrees of freedom $\boldsymbol{d}$ yields

$$\boldsymbol{d} = \boldsymbol{N}_d \boldsymbol{\alpha} = [\boldsymbol{N}_1 | \boldsymbol{N}_2] \left\{ \begin{array}{c} \boldsymbol{\alpha}_1 \\ \boldsymbol{\alpha}_2 \end{array} \right\} \tag{7}$$

where

$$\boldsymbol{N}_d = [\boldsymbol{N}_1 | \boldsymbol{N}_2] = \left[ \begin{array}{cc|ccc} \underset{6\times6}{\boldsymbol{I}} & \underset{6\times6}{\boldsymbol{0}} & \underset{6\times6}{\boldsymbol{0}} & \cdots & \underset{6\times6}{\boldsymbol{0}} \\ \underset{6\times6}{\boldsymbol{I}} & L\underset{6\times6}{\boldsymbol{I}} & L^2\underset{6\times6}{\boldsymbol{I}} & \cdots & L^N\underset{6\times6}{\boldsymbol{I}} \end{array} \right] \tag{8}$$

and

$$\boldsymbol{\alpha} = \left\{ \begin{array}{c} \boldsymbol{\alpha}_1 \\ \boldsymbol{\alpha}_2 \end{array} \right\} = \left[ \begin{array}{c} \underset{12\times12}{\boldsymbol{I}} \\ \underset{(6N-12)\times12}{\boldsymbol{0}} \end{array} \right] \boldsymbol{\alpha}_1 + \left[ \begin{array}{c} \underset{12\times(6N-12)}{\boldsymbol{0}} \\ \underset{(6N-12)\times(6N-12)}{\boldsymbol{I}} \end{array} \right] \boldsymbol{\alpha}_2 = \boldsymbol{A}_1\boldsymbol{\alpha}_1 + \boldsymbol{A}_2\boldsymbol{\alpha}_2 \tag{9}$$

From Equation (7) $\boldsymbol{\alpha}_1$ can be rewritten as

$$\boldsymbol{\alpha}_1 = \boldsymbol{N}_1^{-1}(\boldsymbol{d} - \boldsymbol{N}_2\boldsymbol{\alpha}_2) \tag{10}$$

Substituting (10) into (9) yields

$$\boldsymbol{\alpha} = \underbrace{\boldsymbol{A}_1\boldsymbol{N}_1^{-1}}_{\boldsymbol{Y}_1}\boldsymbol{d} + \underbrace{\left(\boldsymbol{A}_2 - \boldsymbol{A}_1\boldsymbol{N}_1^{-1}\boldsymbol{N}_2\right)}_{\boldsymbol{Y}_2}\boldsymbol{\alpha}_2 \tag{11}$$

The remainder of the the generalized degrees of freedom $\boldsymbol{\alpha}_2$ is obtained by substituting Equation (11) into the elastic energy
(6), and minimizing with respect to $\boldsymbol{\alpha}_2$, which yields

$$\frac{dU}{d\boldsymbol{\alpha}_2} = \underbrace{\boldsymbol{Y}_2^T\boldsymbol{D}\boldsymbol{Y}_1}_{\boldsymbol{P}}\boldsymbol{d} + \underbrace{\boldsymbol{Y}_2^T\boldsymbol{D}\boldsymbol{Y}_2}_{-\boldsymbol{Q}}\boldsymbol{\alpha}_2 = 0 \qquad \Rightarrow \qquad \boldsymbol{\alpha}_2 = \boldsymbol{Q}^{-1}\boldsymbol{P}\boldsymbol{d} \tag{12}$$

Substituting Equation (12) into (11) provides

$$\boldsymbol{\alpha} = \underbrace{\left(\boldsymbol{Y}_1 + \boldsymbol{Y}_2\boldsymbol{Q}^{-1}\boldsymbol{P}\right)}_{\boldsymbol{N}_\alpha}\boldsymbol{d} \tag{13}$$

which allows to express the elastic energy (6) with respect to the nodal degrees of freedom

$$U = \frac{1}{2}\boldsymbol{d}^T\underbrace{\boldsymbol{N}_\alpha^T\boldsymbol{D}\boldsymbol{N}_\alpha}_{\boldsymbol{K}_{el}}\boldsymbol{d} \tag{14}$$

where $\boldsymbol{K}_{el}$ is the element stiffness matrix with respect to the nodal degrees of freedom.



A consistent mass matrix of the element $\boldsymbol{M}_{el}$ is obtained from the kinetic energy

$$T = \frac{1}{2}\dot{\boldsymbol{d}}^T \boldsymbol{N}_\alpha^T \underbrace{\int\limits_{z=0}^{L} \boldsymbol{N}^T \boldsymbol{M}_{cs} \boldsymbol{N} \, dz}_{\boldsymbol{M}_{el}} \boldsymbol{N}_\alpha \dot{\boldsymbol{d}} \tag{15}$$

where $\boldsymbol{M}_{cs}$ is the cross-sectional mass matrix.

## 2.5 Validation

The implementation of the anisotropic beam element into the aero-servo-elastic analysis tool HAWCStab2 has been validated against various test cases of previous publications and by comparison of eigenfrequencies and steady state results of the DTU 10 MW RWT with flap-twist coupled blades.

### 2.5.1 Eigenfrequencies of a coupled cantilever

Hodges et al. (1991) present the natural frequencies of a coupled cantilever box beam. The beam is $2.54$ m long, has a height of $16.76$ mm ($0.66$ in) and a width of $33.53$ mm ($1.32$ in). The wall thickness is $0.84$ mm ($0.033$ in) with six layers of unidirectional lamina stacked $(20/-70/20/-70/-70/20)$ from outside to inside. The material is T 300 / 5208 Graphite / Epoxy with properties provided by Stemple and Lee (1988). The material density is given by Hodges et al. as $1604$ kg/m$^3$ ($1.501 \cdot 10^{-4}$ lbsec$^2$/in$^4$). The cross-section stiffness matrix was taken from Hodges et al. and converted to SI units

$$\boldsymbol{K}_{cs} = \begin{bmatrix} 5.0576 \cdot 10^6 & 0 & 0 & -1.7196 \cdot 10^4 & 0 & 0 \\ & 7.7444 \cdot 10^5 & 0 & 0 & 8.3270 \cdot 10^3 & 0 \\ & & 2.9558 \cdot 10^5 & 0 & 0 & 9.0670 \cdot 10^3 \\ & & & 1.5041 \cdot 10^2 & 0 & 0 \\ & \text{sym.} & & & 2.4577 \cdot 10^2 & 0 \\ & & & & & 7.4529 \cdot 10^2 \end{bmatrix} \tag{16}$$

The cantilever was discretised with 16 elements. Table 1 shows a comparison of the frequencies obtained with the present beam model, the beam models by Hodges et al. (1991) and Armanios and Badir (1995), and a finite element shell model by Kim et al. (2013).

### 2.5.2 Tip displacements and rotations of a coupled cantilever

Wang et al. (2014) present a coupled cantilever beam with a tip load. The stiffness matrix in the original study is

$$\boldsymbol{K}_{cs} = \begin{bmatrix} 1368.17 & 0 & 0 & 0 & 0 & 0 \\ & 88.56 & 0 & 0 & 0 & 0 \\ & & 38.78 & 0 & 0 & 0 \\ & & & 16.96 & 17.61 & -0.351 \\ & \text{sym.} & & & 59.12 & -0.370 \\ & & & & & 141.47 \end{bmatrix} \cdot 10^3 \tag{17}$$

The beam has a length of 10 m and was discretised by 10 elements. A tip load of 150 N was applied to the cantilever. The tip displacements and rotations (in Wiener-Milenkovic Parameter) are shown in Table 2.




| Mode | Freq. [Hz] | | | | Rel. Diff. [%] | | |
|---|---|---|---|---|---|---|---|
| | Present | Hodges | Armanios | Kim | Hodges | Armanios | Kim |
| 1 vert. | 2.99 | 3.00 | 2.96 | 2.98 | 0.3 | -1.0 | -0.3 |
| 1 horiz. | 5.18 | 5.19 | 5.10 | 5.12 | 0.2 | -1.5 | -1.2 |
| 2 vert. | 18.75 | 19.04 | 18.54 | 18.65 | 1.5 | -1.1 | -0.5 |
| 2 horiz. | 32.36 | 32.88 | 31.98 | 32.02 | 1.6 | -1.2 | -1.1 |
| 3 vert. | 52.44 | 54.65 | 51.92 | 52.17 | 4.2 | -1.0 | -0.5 |
| 3 horiz. | 89.40 | 93.39 | 89.55 | 93.39 | 4.5 | 0.2 | 4.5 |
| 1 tors. | 180.10 | 180.32 | 177.05 | - | 0.1 | -1.7 | - |
| 2 tors. | 542.05 | 544.47 | 531.15 | - | 0.4 | -2.0 | - |

**Table 1.** Eigenfrequencies of a coupled cantilever obtained with the present model compared to results by Hodges et al. (1991), Armanios and Badir (1995) (both beam models) and Kim et al. (2013) (FEM model).

| | $u_1$ | $u_2$ | $u_3$ | $\theta_1$ | $\theta_2$ | $\theta_3$ |
|---|---|---|---|---|---|---|
| Present | -0.0902 | -0.0651 | 1.2300 | 0.1845 | -0.1799 | 0.0049 |
| Wang | -0.0906 | -0.0648 | 1.2300 | 0.1845 | -0.1799 | 0.0049 |
| Rel. Diff. [%] | -0.4744 | 0.3472 | 0.0049 | 0.0275 | 0.0124 | 0.1709 |

**Table 2.** Tip displacements and rotations (in Wiener-Milenkovic Parameter) of a coupled cantilever obtained with the present model compared to results by Wang et al. (2014).

### 2.5.3 Bend cantilever

Bathe and Bolourchi (1979) present the geometric nonlinear response of a 45° bend cantilever with a radius of 100 m as shown in Figure 1. The test case has been extended with bend-twist coupled cross-sectional properties by Stäblein and Hansen (2016). A square unit cross section with a modulus of elasticity of $1.0 \cdot 10^7$ N/m² was used for the analysis. Bend-twist coupling was

5   introduced by setting $K_{46} = -0.3\sqrt{K_{44}K_{66}}$ of the cross-section stiffness matrix. A tip load of 300 N has been applied. Table 3 shows the tip displacement of the uncoupled beam compared to results by Simo and Vu_Quoc (1986). And the coupled beam

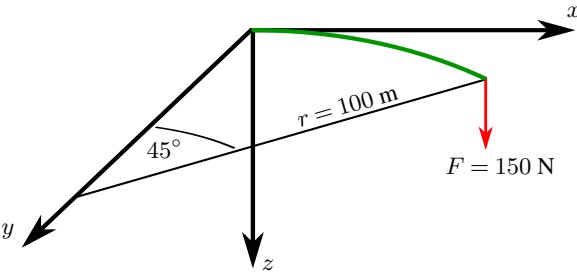

**Figure 1.** 45° bend cantilever.



compared to results of a Timoshenko beam element with anisotropic cross-sectional properties by Stäblein and Hansen (2016).

### 2.5.4 Eigenfrequencies and steady state results for DTU 10 MW RWT blade

The anisotropic beam element by Kim et al. (2013) has previously been implemented in HAWC2, an aeroelastic time-domain
analysis tool for wind turbines capable of computing structural modal properties at standstill. The HAWCStab2 implementa-
tion was therefore compared to HAWC2 by analysing the natural frequencies at standstill, and steady-state power and thrust
(ignoring wind shear, yaw, tilt, turbulence, tower shadow and gravity) of a bend-twist to feather coupled blade for the DTU
10 MW RWT. The blade is coupled with a constant coefficient of $\gamma_y = -0.2$ along the blade. For the comparison, only the
cross-sectional properties of the blade were modified. The twist distribution and pitch angle were adopted from the reference
turbine which explains the unusual shape of the power curve. The first ten natural frequencies are compared in Table 4 and the
results show only minimal differences. The power and thrust over the operational wind speed range are compared in Figure 2
and, again, the results show only minimal differences.

## 3 Bend-Twist Coupled DTU 10 MW Blade

The effects of bend-twist coupling on the modal properties and stability of wind turbine blades were investigated with the DTU
10 MW Reference Wind Turbine (RWT) developed by Bak et al. (2013). It is a horizontal axis, variable pitch, variable speed
wind turbine with a rotor diameter of 178 m and a hub height of 119 m. The structural properties of the blades in terms of $6 \times 6$
cross-section stiffness matrices were obtained with BECAS (Blasques, 2011) and the input data provided on the DTU 10 MW
RWT project homepage[1]. Bend-twist coupling was introduced by setting entries $K_{46}$, which couples flapwise bending with

---

[1]http://dtu-10mw-rwt.vindenergi.dtu.dk

| | Displacement [m] | | | Rel. Diff. [%] | | |
|---|---|---|---|---|---|---|
| | x | y | z | x | y | z |
| Simo & Vu-Quoc | -11.87 | -6.96 | 40.08 | - | - | - |
| Present uncoupled | -12.15 | -7.17 | 40.48 | 2.3 | 3.1 | 1.0 |
| Stäblein & Hansen | -10.66 | -6.53 | 38.68 | - | - | - |
| Present coupled | -10.65 | -6.56 | 38.69 | -0.1 | 0.4 | 0.0 |

**Table 3.** Comparison of $45°$ bend cantilever tip displacements. Original test case (uncoupled) and modified test case with bend-twist coupling.




| Mode # | HAWCStab2 [Hz] | HAWC2 [Hz] | Abs. Diff. [Hz] |
|---|---|---|---|
| 1 | 0.59167 | 0.59167 | 0.00000 |
| 2 | 0.92880 | 0.92880 | 0.00000 |
| 3 | 1.69030 | 1.69030 | 0.00000 |
| 4 | 2.75012 | 2.75012 | 0.00000 |
| 5 | 3.48110 | 3.48110 | 0.00000 |
| 6 | 5.61503 | 5.61502 | 0.00001 |
| 7 | 6.02067 | 6.02065 | 0.00002 |
| 8 | 6.70144 | 6.70144 | 0.00000 |
| 9 | 8.72890 | 8.72884 | 0.00006 |
| 10 | 9.86475 | 9.86472 | 0.00003 |

**Table 4.** Natural frequency comparison of a flap-twist to feather coupled DTU 10 MW RWT blade ($\gamma_y = -0.2$) obtained with HAWCStab2 and HAWC2.

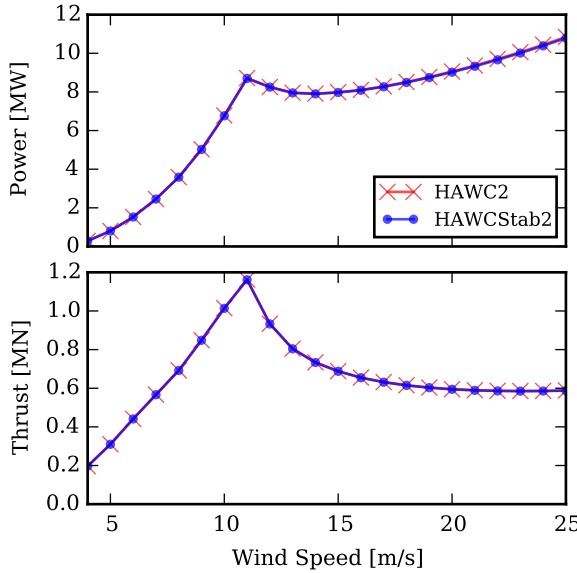

**Figure 2.** Power and thrust of the DTU 10 MW RWT with flap-twist to feather coupled blades ($\gamma_y = 0.2$) obtained with HAWC2 and HAWCStab2. Note that the blades have not been pretwisted and the pitch angel has not been adjusted for this graph.





torsion, and $K_{56}$, coupling edgewise bending with torsion, to

$$K_{46} = \gamma_y \sqrt{K_{44} K_{66}} \qquad (18)$$
$$K_{56} = -\gamma_x \sqrt{K_{55} K_{66}} \qquad (19)$$

where $\gamma_x$ and $\gamma_y$ are edge- and flap-twist coupling coefficients as proposed by Lobitz and Veers (1998), and $K_{44}$, $K_{55}$ and

$K_{66}$ are flapwise bending, edgewise bending and torsional stiffness of the cross-section. For a positive definite stiffness matrix the coupling coefficients have to be $|\gamma_{x/y}| < 1$. For wind turbine blades values up to 0.2-0.4 are deemed achievable (Capellaro and Kühn, 2010; Fedorov and Berggreen, 2014). Negative coupling coefficients result in pitch to feather (reducing the angle of attack) for edgewise/flapwise deflection towards the leading edge/suction side of the blade. Positive coupling coefficients result in pitch to stall (increasing the angle of attack).

To reduce the coupling related power loss the blades were pretwisted at a reference wind speed of 8 m/s using the procedure presented by Stäblein et al. (2016b). Figure 3 shows the aeroelastic twist along the blade for the reference and flap-twist to feather and stall coupled blades with coupling coefficients $\gamma_y = \pm 0.1$ constant along the blades. The aerodynamic twist of the flap-twist to feather coupled blade increases towards the blade tip, compared to the reference blade, to compensate for the coupling induced twist. The flap-twist to stall coupled blade has a lower aerodynamic twist towards the blade tip. As a result of

the pretwisting procedure, the steady state angle of attack and hence the aerodynamic states are identical for all models at the reference wind speed of 8 m/s. After pretwisting the blade, the pitch angles that optimize power below rated and limit power above rated were recalculated. The pitch angles have a lower bound of $0°$ and are constrained by a maximum angle of attack of $8°$ in the outer part of the blade. The pitch angles over wind speed for the reference and flap-twist coupled blades are shown in Figure 4.

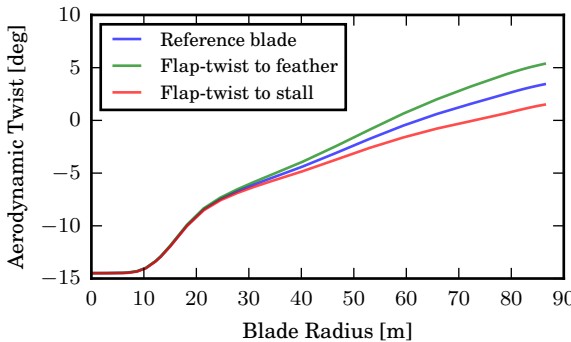

**Figure 3.** Aerodynamic twist along the blade for the reference and flap-twist to feather and stall coupled blades with coupling coefficients $\gamma_y = \pm 0.1$ constant along the blades.





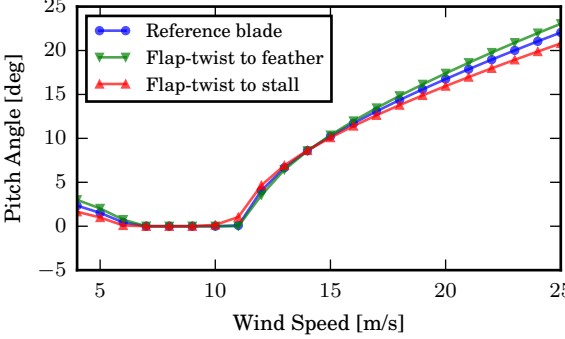

**Figure 4.** Pitch angles over wind speed for the reference and flap-twist to feather and stall coupled blades with coupling coefficients $\gamma_y = \pm 0.1$ constant along the blade.

## 4 Results

In this section, the structural and aeroelastic modal properties of bend-twist coupled and pretwisted blades are investigated. First, the results of blade only analysis are presented followed by some additional investigations where the turbine dynamics have also been considered. The results focuses on the first edgewise and first flapwise blade modes as the effects of bend-twist

coupling on the frequency and damping are most distinct for those mode shapes. Also, the first edgewise mode is the lowest damped blade mode and the first to become unstable.

### 4.1 Blade Modal Properties

First, the effects of coupling on the structural mode shapes of the unloaded blade are investigated. Figure 5 and 6 show the structural mode shapes of the first edgewise mode for edge-twist coupled blades and the first flapwise mode for flap-twist

coupled blades. The edgewise and flapwise amplitudes are similar for all blades. The structural properties and upwind prebend of the reference blade result in a tip twist of $0.5°$ towards stall for the edgewise mode. Edge-twist coupling of the cross-section stiffness matrix results in an additional tip twist of about $0.5°$ towards feather for $\gamma_x = -0.1$, or towards stall for $\gamma_x = +0.1$ relative to the reference blade. The flapwise mode of the reference blade has a tip twist of about $0.3°$ towards feather. Flap-twist coupling results in an additional tip twist of about $0.4°$ towards feather for $\gamma_y = -0.1$, or towards stall for $\gamma_y = +0.1$.

#### 4.1.1 Aeroelastic Frequency and Damping over Coupling Coefficients

The bend-twist coupling also affects the structural and aeroelastic modal frequency and damping. The aeroelastic modal properties are compared at 8 m/s where the aerodynamic steady states (and the gradients around the linearisation point) are the same for all blades because this wind speed is used in the pretwisting of the coupled blades. Figure 7 shows contour plots of the structural (left column) and aeroelastic (middle column) modal frequencies for the first edgewise (top row) and first flapwise

(bottom row) mode. The difference between structural and aeroelastic frequency is also plotted (right column) to show the



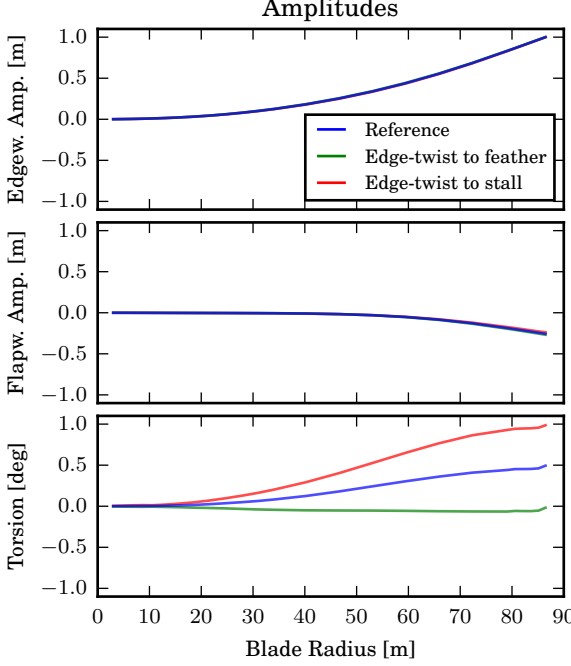

**Figure 5.** Structural mode shape of the first edgewise mode for the reference blade, and edge-twist coupled blades with constant coupling coefficients of $\gamma_x = \pm0.1$. Positive coupling coefficients denote twist to stall for edgewise deflection towards the leading edge. The amplitudes are normalised to 1 m tip deflection in the edgewise direction.

effect of the aerodynamic forces. The contour plots have been obtained with a coupling coefficient step size of 0.01, resulting in a total of 2601 blade models. Each individual blade has been pretwisted to ensure the same angle of attack along the blade as the reference blade. The structural and aeroelastic frequencies of the edgewise mode (top row) are mainly influenced by edge-twist coupling. Flap-twist coupling has only a small influence. The structural frequency has a maximum around $\gamma_x = -0.1$ and

5   reduces from there for both, twist to feather and stall. The frequency difference on the top right shows that the aerodynamic forces increase the frequency for edge-twist to stall coupling and reduce it for edge-twist to feather. The frequencies of the flapwise mode (bottom row) are mainly influenced by flap-twist coupling. The structural frequency has a maximum around $\gamma_y = 0.05$ and reduces from there for both twist to feather and stall. The aerodynamic forces result in a frequency increase for flap-twist to feather and a reduction for flap-twist to stall. The frequency change due to flap-twist coupling is somewhat larger

10   than for edge-twist coupling.

Figure 8 shows contour plots of the structural (left column) and aeroelastic (middle column) modal damping for the first edgewise (top row) and first flapwise (bottom row) mode. The difference between structural and aeroelastic damping is also plotted (right column) to show the effect of the aerodynamic forces. For the edgewise mode, the structural damping contributes about 25% to the aeroelastic damping. The aeroelastic damping tends to increase for edge-twist to feather and reduce for edge-



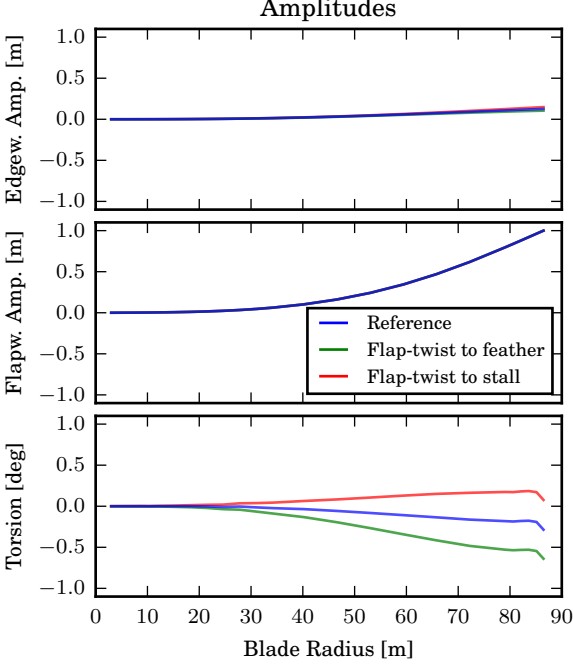

**Figure 6.** Structural mode shape of the first flapwise mode for the reference blade, and flap-twist coupled blades with constant coupling coefficients of $\gamma_y = \pm 0.1$. Positive coupling coefficients denote twist to stall for flapwise deflection towards the suction side. The amplitudes are normalised to 1 m tip deflection in the flapwise direction.

twist to stall coupling. For the flapwise mode, the structural contribution to the aeroelastic damping is negligible. Damping increases for flap-twist to stall coupling and reduces for flap-twist to feather.

### 4.1.2 Aeroelastic Frequency and Damping over Operational Range

The effect of bend-twist coupling on frequencies and damping over the operational range of the turbine has also been inves-
5  tigated. Figure 9 shows aeroelastic frequency (top) and damping ratio (bottom) over the operational wind speed range for the first edgewise blade only mode for the reference, and edge-twist coupled blades with coupling coefficients of $\gamma_x = \pm 0.1$. The frequency of the edgewise mode changes little with the coupling. Damping increases in the wind speed range between 6 and 11 m/s (where the pitch angle is close to zero) for the edge-twist to feather coupled blade and damping reduces for edge-twist to stall coupling. Outside that region where the blade is pitched (cf. Figure 4) damping reduces for edge-twist to feather coupling
10  and increases for edge-twist to stall.

Figure 10 shows aeroelastic frequency (top) and damping ratio (bottom) over the operational wind speed range of the first flapwise blade mode for the reference, and flap-twist coupled blades with coupling coefficients of $\gamma_y = \pm 0.1$. The frequency of the flapwise mode increases over the whole operational range for flap-twist to feather coupling and reduces for flap-twist to stall. Damping reduces over the whole operational range for flap-twist to feather and increases for flap-twist to stall.



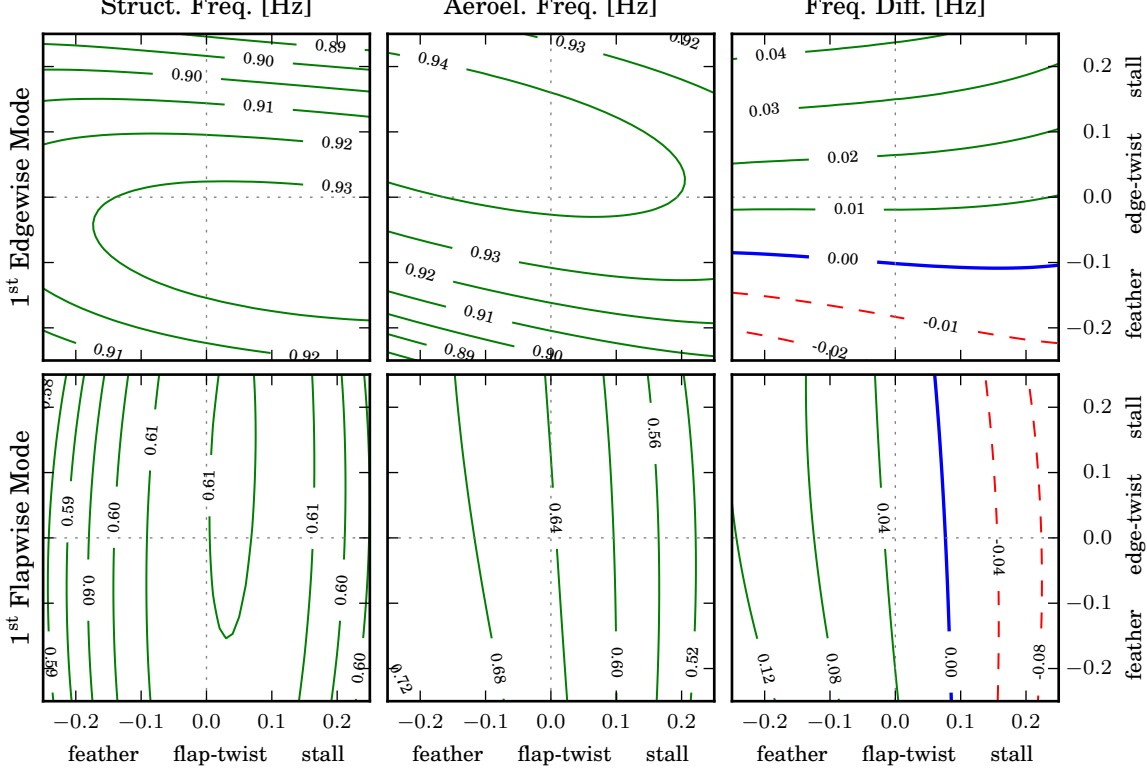

**Figure 7.** Contour plots of structural (left) and aeroelastic (middle) frequencies and their difference (right) of the first edgewise (top) and first flapwise (bottom) mode for varying edge-twist (ordinate) and flap-twist (abscissa) coupling coefficients at 8 m/s wind speed.

The effect of flap-twist coupling on the edgewise mode over the operational range of the turbine has also been examined. Figure 11 shows aeroelastic frequency (top) and damping ration (bottom) over the operational wind speed range of the first edgewise blade mode for the reference, and flap-twist coupled blades with coupling coefficients of $\gamma_y = \pm 0.1$. The frequency of the edgewise mode is not influenced by the flap-twist coupling. Damping for the coupled blades varies around the reference

5  blade but it remains close to the damping of the reference blade.

### 4.1.3 Aeroelastic Mode Shapes

Next, the modes shapes are investigated to identify the cause of the changes in aeroelastic damping. Figure 12 shows the amplitudes and phase angles of the first edgewise aeroelastic mode at 8.0 m/s wind speed for the reference and edge-twist to feather and stall coupled blades with coupling coefficients of $\gamma_x = \pm 0.1$. The amplitudes are normalized to 1.0 m tip deflection

10  in the edgewise direction, and the phase angles are relative to the edgewise tip deflection. The phase angle of the edgewise component is close to zero along the blade. The flapwise components at the tip and phase angles in the outer part are similar for all blades, 0.27 m and $-60°$ for the reference, 0.24 m and $-40°$ for the edge-twist to feather, and 0.33 m and $-70°$ for the




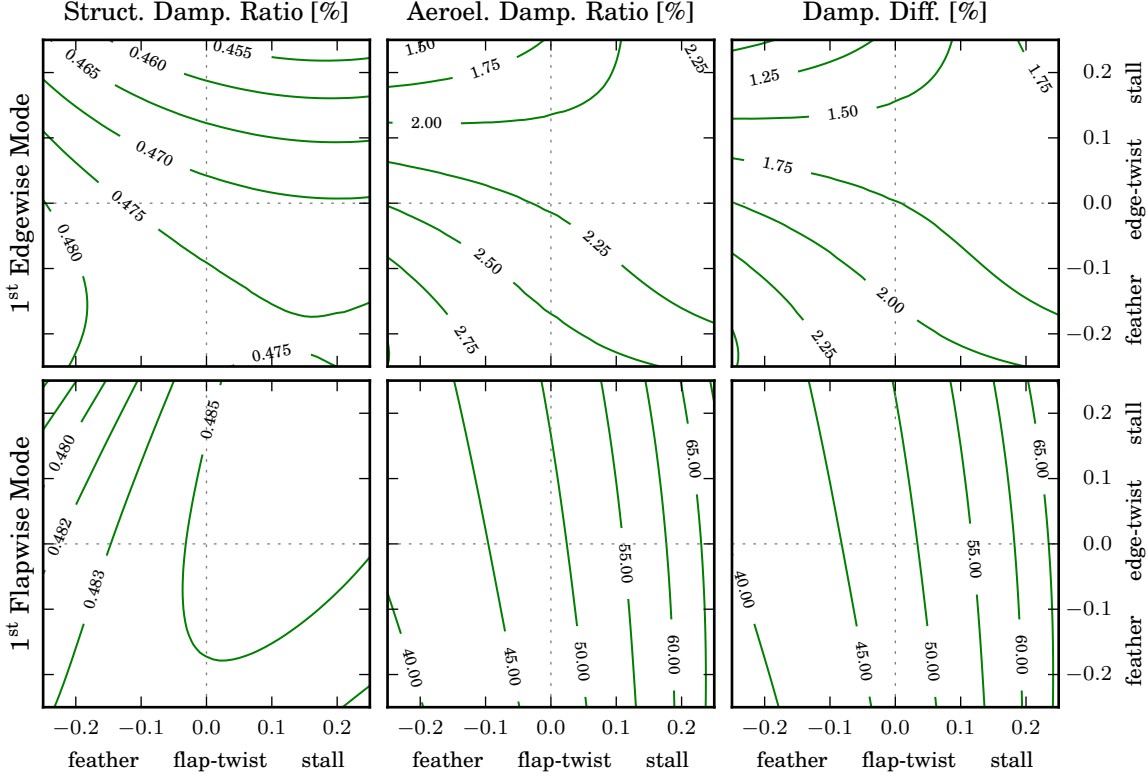

**Figure 8.** Contour plots of structural (left) and aeroelastic (middle) damping ratios and their difference (right) of the first edgewise (top) and first flapwise (bottom) mode for varying edge-twist (ordinate) and flap-twist (abscissa) coupling coefficients at 8 m/s wind speed.

edge-twist to stall coupled blade. The torsional component at the tip of the reference blade is $0.18°$ and has a phase angle of $170°$. Thus, in contrast to the structural mode shape, the reference blade tip twists towards feather (instead of stall) for edgewise deflection towards the leading edge. This twist to feather coupling is caused by the nonlinear geometric coupling when the blade is bend downwind due to the mean aerodynamic forces. The torsional components and phase angles of the coupled blades are

5    $0.68°$ and $-180°$ for edge-twist to feather, and $0.37°$ and $15°$ for edge-twist to stall. In an earlier publication (Stäblein et al., 2016a) it has been shown that edgewise damping is dominated by the work of the lift which reduces when the lift is ahead of the flapwise component. As the flapwise component of the coupled blades are in the same order, it is sufficient to focus on the difference in amplitude and phase angle of the torsional component. For the edge-twist to feather coupled blade, torsion is lagging $140°$ behind the flapwise component (i.e. the lift induced by torsion is almost in counterphase with the flapwise

10    velocity) which increases the work of the lift force and the damping. The reference blade has a similar phase angle but a lower torsional amplitude resulting in lower damping compared to the edge-twist to feather coupled blade. For the edge-twist to stall coupled blade, torsion is $85°$ ahead of the flapwise component and the work of the lift (and the damping) is reduced.





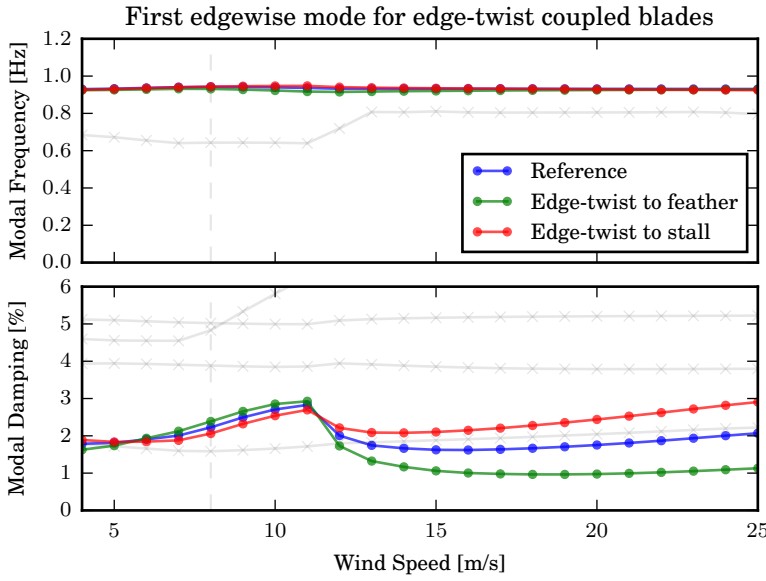

**Figure 9.** Aeroelastic frequency (top) and damping ratio (bottom) over the operational wind speed range of the first edgewise blade only mode for the reference and edge-twist coupled blades with coupling coefficients of $\gamma_x = \pm 0.1$. The grey solid lines indicate other blade modes of the reference blade. The grey dashed line indicates the pretwisting reference speed of 8 m/s.

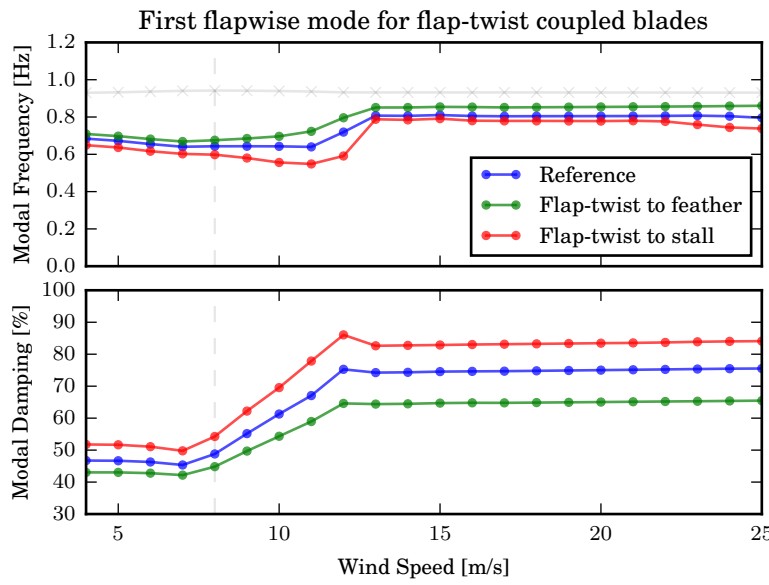

**Figure 10.** Aeroelastic frequency (top) and damping ratio (bottom) over the operational wind speed range of the first flapwise blade only mode for the reference and flap-twist coupled blades with coupling coefficients $\gamma_y = \pm 0.1$. The grey solid lines indicate other blade modes of the reference blade. The grey dashed line indicates the pretwisting reference speed of 8 m/s.



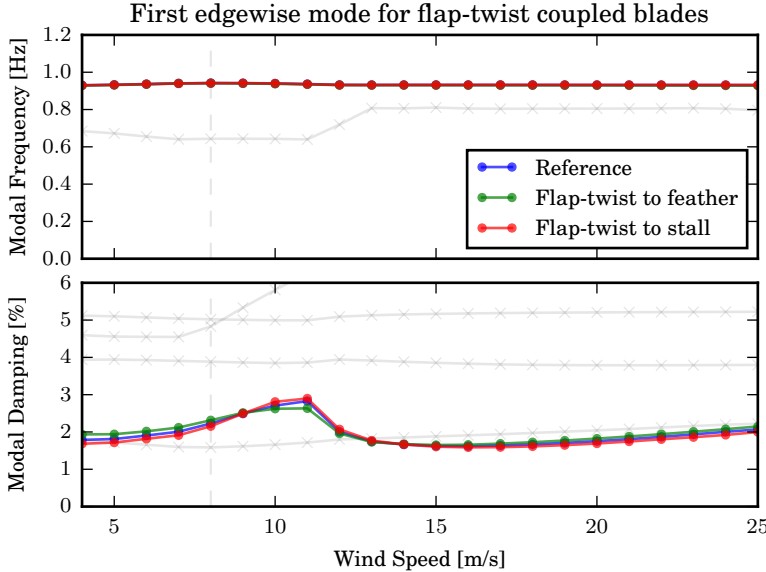

**Figure 11.** Aeroelastic frequency (top) and damping ratio (bottom) over the operational wind speed range of the first edgewise blade only mode for the reference and flap-twist coupled blades with coupling coefficients $\gamma_y = \pm 0.1$. The grey solid lines indicate other blade modes of the reference blade. The grey dashed line indicates the pretwisting reference speed of 8 m/s.

Figure 13 shows the amplitudes and phase angles of the first edgewise aeroelastic mode at 16.0 m/s wind speed for the reference and edge-twist to feather and stall coupled blades with coupling coefficients of $\gamma_x = \pm 0.1$. The flapwise components of the reference and edge-twist to feather coupled blade reduce compared to the edgewise mode at 8 m/s wind speed. The flapwise components at the tip and phase angles in the outer part are, 0.13 m and $-60°$ for the reference, 0.07 m and $5°$

for the edge-twist to feather, and 0.28 m and $-80°$ for the edge-twist to stall coupled blade. The torsional components and phase angles are $0.09°$ and $-130°$ for the reference, $0.55°$ and $-170°$ for the edge-twist to feather, and $0.52°$ and $0°$ for the edge-twist to stall coupled blade. The lower damping for edge-twist to feather compared to edge-twist to stall coupled blades can be explained by the flapwise amplitude which is three times larger for the edge-twist to stall coupled blade. The increased amplitude results in a larger damping due to the direct coupling of the angle of attack/lift and the flapwise velocity.

The torsional component of the edge-twist to feather coupled blade is close to counterphase with the flapwise velocities and therefore has little influence on the damping. The torsional component of the edge-twist to stall coupled blade is nearly in phase with the flapwise velocities which reduces the damping.

Figure 14 shows the amplitudes and phase angles of the first flapwise mode at 8.0 m/s wind speed for the reference and flap-twist to feather and stall coupled blades with coupling coefficients of $\gamma_y = \pm 0.1$. The amplitudes are normalized to 1.0 m

tip deflection in the flapwise direction, and the phase angles are relative to the flapwise tip deflection. The phase angle of the flapwise component is about $10°$ in the outer part of the blade. The edgewise components at the tip are around 0.17 m and in phase with the flapwise blade tip deflections for all three blades. The torsional components and phase angles are $0.36°$ and

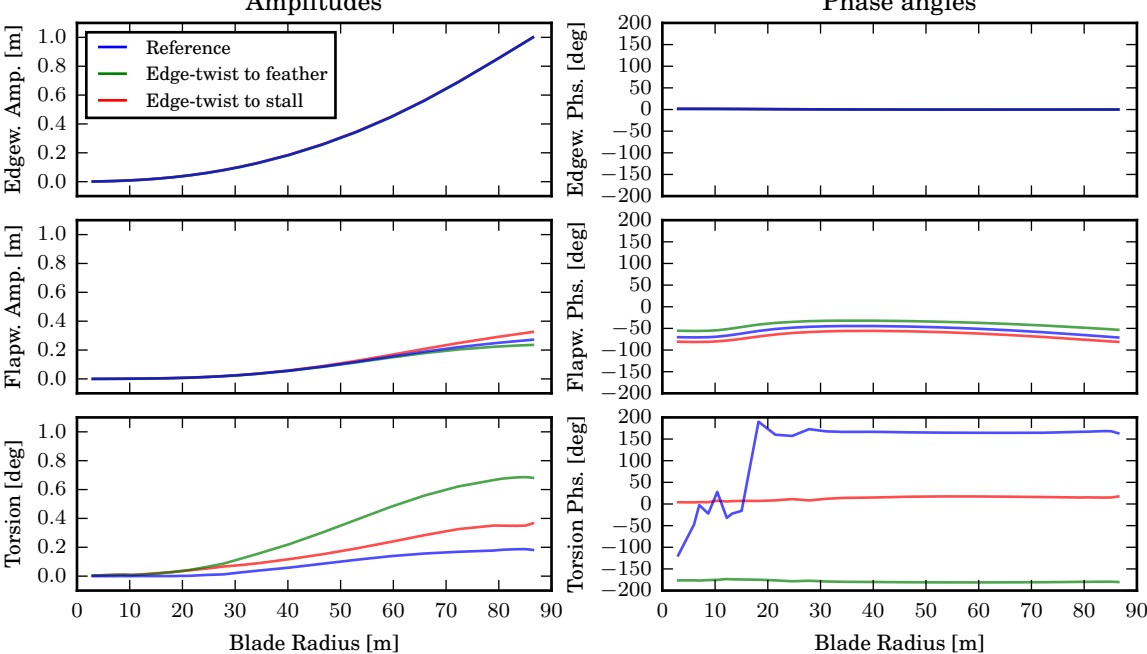

**Figure 12.** Amplitudes (top) and phase angles (bottom) of the first aeroelastic edgewise mode at 8 m/s wind speed for the reference (blue), and edge-twist to feather (green) and stall (red) coupled DTU 10 MW RWT blades. Coupling coefficients are $\gamma_x = \pm 0.1$ constant along the blade. Amplitudes are normalized to 1.0 m tip deflection in the edgewise direction, and phase angles are relative to the edgewise tip deflection.

$-120°$ for the reference, $0.68°$ and $-160°$ for the edge-twist to feather, and $0.40°$ and $-35°$ for the edge-twist to stall coupled blade. As for the edgewise mode, the flapwise damping is dominated by the work of the lift. For the flap-twist to feather coupled blade torsion is close to counterphase with the flapwise component resulting in a torsional component that contributes little to the work of the lift and the damping. The torsional phase angle of the flap-twist to stall coupled blade on the other hand is lagging the flapwise component by about $45°$ which results in an increased work of the lift. Together with the reduced frequency the increased lift work results in higher damping.

## 4.2 Runaway Analysis

The stability of bend-twist coupled blades has been investigated in a runaway scenario where the wind speed is slowly increased while the pitch angle is set to $0°$ and the generator torque is zero. The stability analysis has been conducted with four different coupling coefficients $\gamma_x = \pm 0.1$ and $\gamma_y = \pm 0.1$ along the whole blade span. Figure 15 shows aeroelastic frequency and damping over the tip speed for the lowest damped mode which is the first edgewise mode in such a runaway scenario. The eigenvalue analysis shows that the reference blade becomes unstable at a tip speed of about 180 m/s. Bak et al. (2013) report an edgewise instability at approximately 22 rpm or 205 m/s at the tip using nonlinear time domain analysis. The edge-twist to

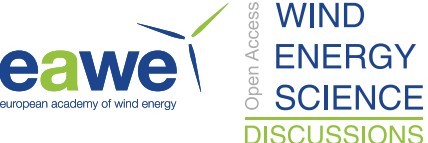

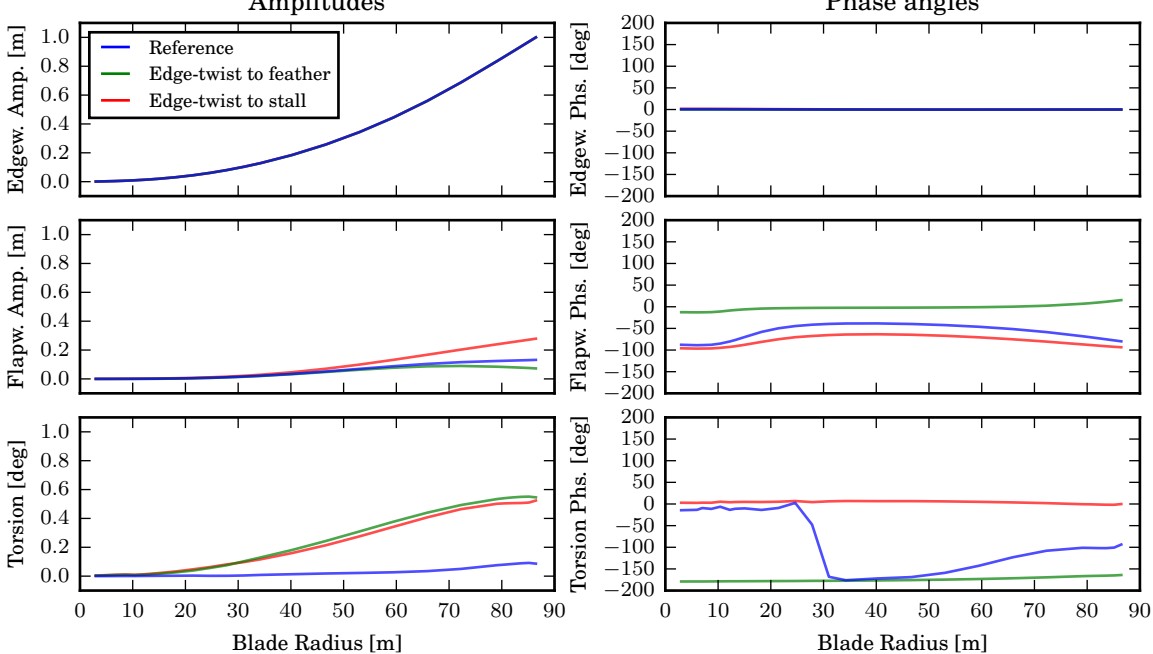

**Figure 13.** Amplitudes (top) and phase angles (bottom) of the first aeroelastic edgewise mode at 16 m/s wind speed for the reference (blue), and edge-twist to feather (green) and stall (red) coupled DTU 10 MW RWT blades. Coupling coefficients are $\gamma_x = \pm 0.1$ constant along the blade. Amplitudes are normalized to 1.0 m tip deflection in the edgewise direction, and phase angles are relative to the edgewise tip deflection.

feather coupled blade has a slightly lower frequency and a lower damping than the reference blade. The lower damping of the edge to feather coupled blade results in instability at a much lower tip speed of 130 m/s. The edge to stall coupled blade shows a slightly higher frequency and a higher damping than the uncoupled blade. Frequency and damping of the flap-twist to feather coupled blade is very close to the reference. The flap-twist to stall coupled blade is close to the reference until a tip speed of

5 about 140 m/s where the frequency starts to reduce and, as a result, damping to increase. This behaviour is characteristic for a mode that approaches divergence instability.

  The mode shapes of the reference and edge-twist coupled blades at a tip speed of about 140 m/s of the runaway scenario are shown in Figure 16. The amplitudes are normalized to 1.0 m tip deflection in the edgewise direction, and the phase angles are relative to the edgewise tip deflection. The phase angle of the edgewise component is close to zero along the blade. The

10 flapwise components at the tip and phase angles in the outer part are, 0.06 m and 55° for the reference, 0.29 m and 75° for the edge-twist to feather, and 0.18 m and −80° for the edge-twist to stall coupled blade. The torsional components and phase angles are 0.46° and −160° for the reference, 0.93° and −170° for the edge-twist to feather, and 0.21° and −35° for the edge-twist to stall coupled blade. As for the previous mode shapes, the difference in damping can be explained by observing the amplitudes and phase angles between the torsional and flapwise components. The edge-twist to feather coupled blade has





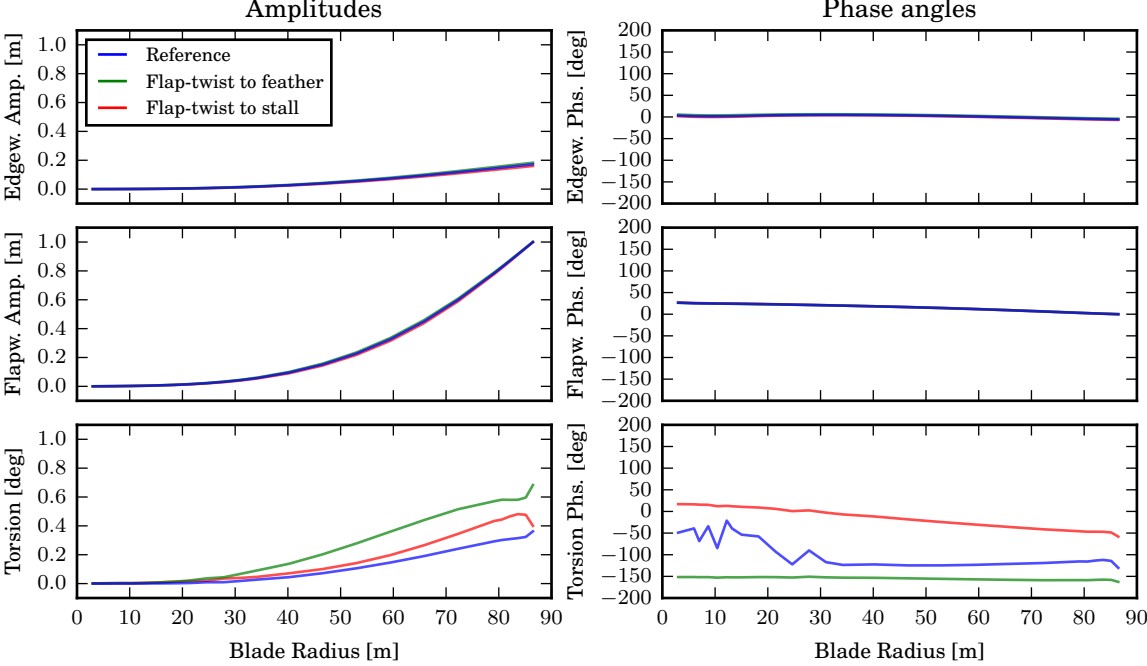

**Figure 14.** Amplitudes (top) and phase angles (bottom) of the first aeroelastic flapwise mode at 8 m/s wind speed for the reference (blue), and flap-twist to feather (green) and stall (red) coupled DTU 10 MW RWT blade. Coupling coefficient is $\gamma_y = \pm 0.1$ constant along the blade. Amplitudes are normalized to 1.0 m tip deflection in the flapwise direction, and phase angles are relative to the flapwise tip deflection.

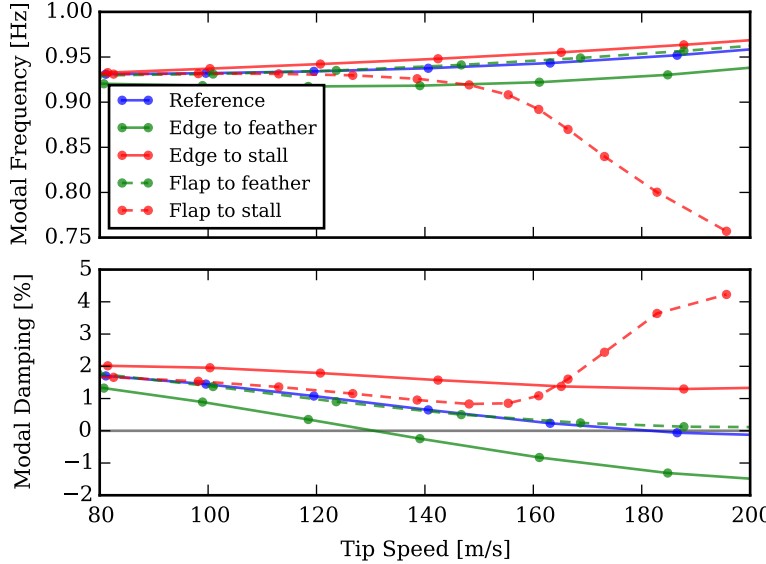

**Figure 15.** Campbell diagram of modal frequency (top) and damping (bottom) for the first edgewise mode in a runaway scenario. Coupling coefficients are $\gamma_{x/y} = \pm 0.1$ constant along the blade.





the largest torsional amplitude and a phase angle that is closest to the flapwise velocities (which are $90°$ ahead of the flapwise component) which reduces the damping. For the reference and edge-twist to feather coupled blades the torsional amplitudes decrease and the phase angle moves away from the flapwise velocity which has a positive effect on the modal damping.

### 4.3  Turbine Modal Properties

The blade only analysis has shown that the damping of the edgewise mode is sensitive to the pitch angle (cf. Figure 12) because the pitching affects the mode shape relative to the inflow. For a stability analysis of the edgewise mode, all factors that could influence the mode shape should therefore be considered. The effect of turbine dynamics on the modal properties of the edgewise mode has been investigated in Figure 17. The plot shows modal frequency and damping of the backward whirling (BW), forward (FW), and symmetric edgewise modes for the reference and edge-twist coupled blades ($\gamma_x = \pm 0.1$).
The grey lines indicate the remaining turbine modes of the reference blade, which counting from low to high frequency are: tower side-side, tower for-aft, and the first and second backward whirling, symmetric, and forward whirling flapwise modes. The frequency of the edgewise modes changes little with the coupling. Between 6 and 11 m/s wind speed damping of the edge-twist to feather coupled blade increases for all three edgewise turbine modes (backward, and forward whirling, and symmetric) as predicted by the blade only analysis. For edge-twist to stall the damping reduces. Above rated wind speed, the damping of
the backward and forward whirling modes drop for all blades. Damping of the symmetric modes increases. Above rated wind speed, edge-twist to feather coupling reduces the damping of the symmetric and backward whirling modes. The damping of the backward whirling mode is very similar to the aeroelastic camping of the edgewise blade mode (cf. Figure 9) and for the edge-twist to feather coupled blade the backward whirling mode becomes the lowest damped mode.

Flap-twist to feather coupled blades have been reported to reduce fatigue loads of the flapwise blade root bending moment
of the blade (Lobitz et al., 1999; Lobitz and Veers, 2003; Verelst and Larsen, 2010; Bottasso et al., 2013). To investigate the load alleviation in frequency domain, the frequency response of the flapwise blade root bending moment to mean wind speed variation between 0 Hz and 2 Hz for steady state operation at mean wind speeds of 5, 10, 15, and 20 m/s are shown in Figure 18. The coupling coefficient is $\gamma_y = \pm 0.1$ constant along the blade. The flap-twist to feather coupled blade shows a reduced magnitude for wind speed variations below 0.5 Hz. The magnitude increases for flap-twist to stall coupled blades. Above
0.5 Hz the frequency response is similar for all blades.

Figure 19 shows the frequency response of the tower bottom for-aft moment to mean wind speed variation between 0 Hz and 0.5 Hz for steady state operation at mean wind speeds of 5, 10, 15, and 20 m/s. Flap-twist to feather coupling tends to reduce the frequency response for all operational points while the response increases for flap-twist to stall coupled blades.

### 5  Discussion

Edge-twist coupling has only a small influence on the frequency of the edgewise mode. Damping increases for edge-twist to feather coupling when the pitch angle is close to zero (cf. Figure 4) and reduces for edge-twist to stall coupling respectively. For wind speeds where the blade is pitched, the damping of the edgewise mode reduces for edge-twist to feather and increases





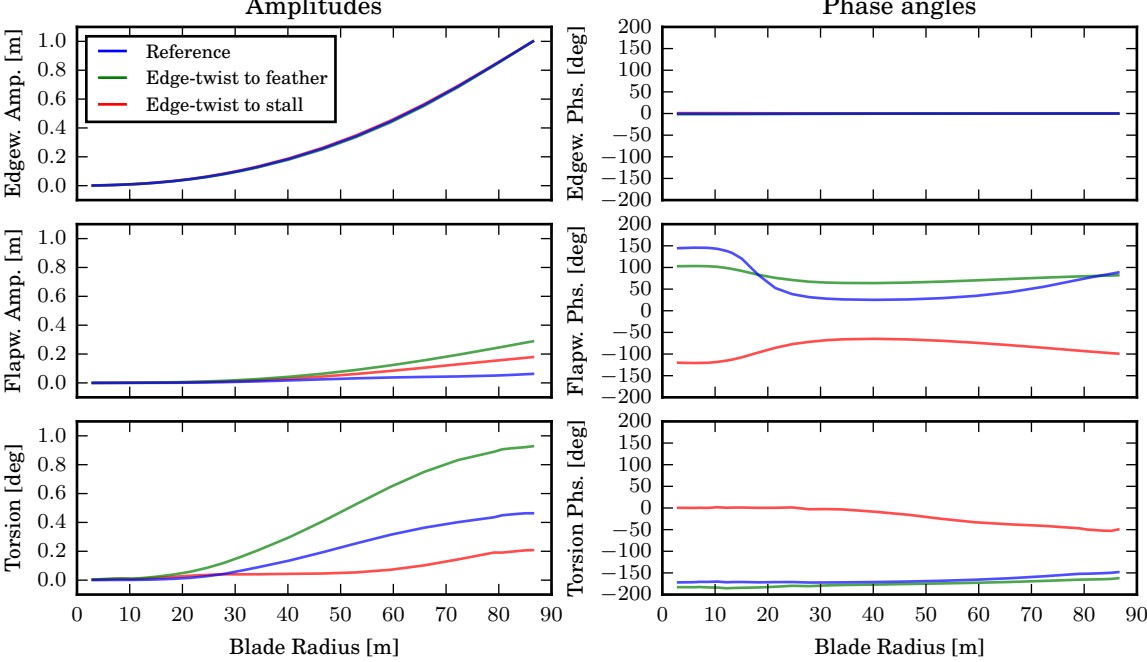

**Figure 16.** Amplitudes (top) and phase angles (bottom) of the first aeroelastic edgewise mode as indicated in Figure 15 at approx. 140 m/s tip speed for the reference (blue), and edge-twist to feather (green) and stall (red) coupled DTU 10 MW RWT blade. Coupling coefficients are $\gamma_x = \pm 0.1$ constant along the blade. Amplitudes are normalized to 1.0 m tip deflection in the edgewise direction, and phase angles are relative to the edgewise tip deflection.

for edge-twist to stall coupling. Increased damping for edge-twist to feather and reduced damping for edge-twist to stall coupling is also observed by Hong and Chopra (1985) and Stäblein et al. (2016a). Rasmussen et al. (1999) on the other hand observe reduced damping for edge-twist to feather and increased damping for edge-twist to stall coupling if the direction of the edgewise vibration is between the inflow and the rotor plane. The qualitative differences of the edge-twist coupling effect on

5   damping reported in previous studies, and the observed change over the operational wind speed range (cf. Figure 9) show that damping of the edgewise mode can be sensitive to changes in the mode shape. The effect of turbine dynamics have therefore been investigated (cf. Figure 17). The results show that the effects of edge-twist coupling on the edgewise turbine modes are similar to the blade only mode (i.e. edge-twist to feather coupling increases damping if the pitch angle is close to zero and reduces damping if the blade is pitched). Analysis of the edgewise mode shape further shows that geometric coupling due to

10   prebending and load deflection has a significant influence on the edgewise mode shape. An observation that has also been made by Kallesøe and Hansen (2009).

    The DTU 10 MW RWT blade becomes unstable due to flutter of the edgewise mode. Edge-twist to feather coupling reduces the critical inflow speed due to an increase of the torsional component of the edgewise mode, and a torsional phase angle that is close to the flapwise velocity. The critical inflow speed increases for edge-twist to stall coupled blades. The formation of



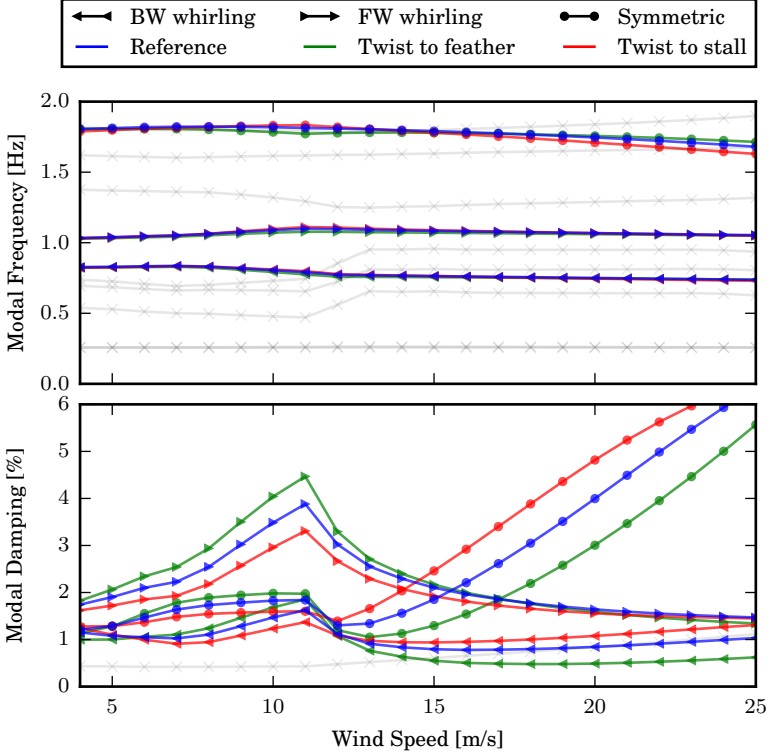

**Figure 17.** Aeroelastic frequency (top) and damping ratio (bottom) over the operational wind speed range of the first edgewise turbine modes for reference and edge-twist coupled blades with coupling coefficients $\gamma_x = \pm 0.1$. The grey lines indicate the remaining turbine modes (tower side-side, fore-aft, first flap bwd., sym., fwd., secong flap bwd., sym., fwd. from low to high frequency) of the reference blade.

an edge-twist flutter mode, where the torsional component of the edgewise mode becomes large enough and in phase with the flapwise velocity has previously been reported by Kallesøe and Hansen (2009) and Stäblein et al. (2016a).

Flap-twist to feather coupling increases the frequency and reduces the damping of the first flapwise blade mode (cf. Figure 10). Flap-twist to stall coupling reduces the frequency and increases the damping respectively. Similar observations have been 5 made in previous studies (Hong and Chopra, 1985; Rasmussen et al., 1999; Hansen, 2011; Stäblein et al., 2016a). Flap-twist coupling has little influence on the damping of the edgewise mode (cf. Figure 11). A similar observation has been made by Hansen (2011) for swept blades. The reduced frequency response to mean wind speed variations of the blade root flapwise moment (cf. Figure 18) concurs with reduced fatigue loads observed by Lobitz et al. (1999); Lobitz and Veers (2003); Verelst and Larsen (2010) and Bottasso et al. (2013). Flap-twist to feather coupling also reduces the tower bottom fore-aft moment (cf. 10 Figure 19).

The inflow speed at which the DTU 10 MW RWT becomes unstable due to flutter of the edgewise mode changes little for flap-twist to feather coupling and increases for flap-twist to stall coupling. The effect of flap-twist coupling on the classical





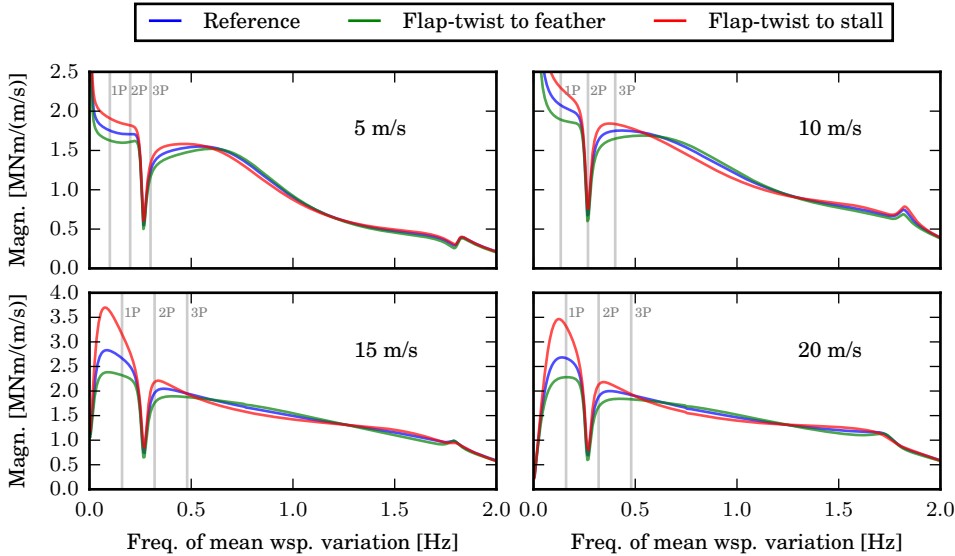

**Figure 18.** Frequency response of the flapwise blade root bending moment to mean wind speed variation between 0.0 Hz and 2.0 Hz for steady state operation at mean wind speeds of 5, 10, 15, and 20 m/s. The coupled blades have a constant coefficient of $\gamma_y = \pm 0.1$.

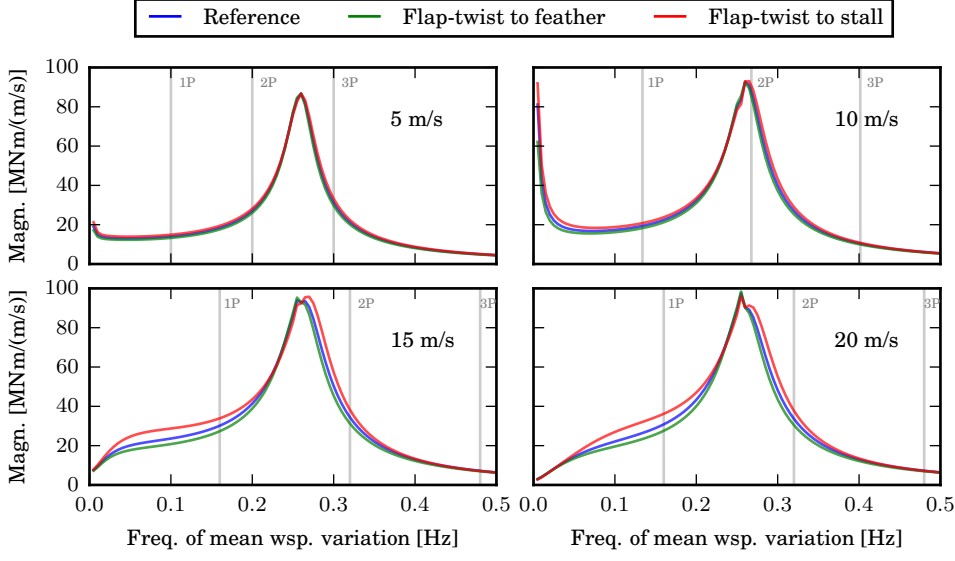

**Figure 19.** Frequency response of the tower bottom fore-aft moment to mean wind speed variation between 0.0 Hz and 0.5 Hz for steady state operation at mean wind speeds of 5, 10, 15, and 20 m/s. The coupled blades have a constant coefficient of $\gamma_y = \pm 0.1$.



flutter (where flapwise and torsional mode coalesce into an unstable mode) and divergence speeds could not be investigated as the first edgewise blade mode of the DTU 10 MW RWT becomes unstable before those speeds are reached.

## 6   Conclusions

In this paper the aeroelastic modal properties and stability limits of the DTU 10 MW RWT with bend-twist coupled blades have been investigated. Coupling has been introduced in the cross-section stiffness matrix by means of coupling coefficients. The aeroelastic modal properties and stability limits of both, edge- and flap-twist coupled blades have been investigated by means of eigenvalue analysis around a steady-state equilibrium using the aero-servo-elastic tool HAWCStab2. For the analysis with fully coupled cross-section stiffness matrices, an anisotropic beam element has been implemented in HAWCStab2 and validated against previously published test cases.

The damping of the first edgewise mode increases for edge-twist to feather coupling, and reduces for edge-twist to stall coupling, if the pitch angle is close to zero. Outside that region, where the blade is pitched, damping reduces for edge-twist to feather and increases for edge-twist to stall coupled blades. The effect of edge-twist coupling on the edgewise turbine modes (forward and backward whirling, and symmetric) is similar to the blade only mode. Analysis of the edgewise mode shows that geometric coupling due to prebending and load deflection has a significant effect on the torsional component of the edgewise mode shape. Edge-twist to feather coupling reduces the critical inflow speed of the turbine due to an increase in the torsional component, and a torsional phase angle that approaches the flapwise velocity.

The results on flap-twist coupled blades confirm the findings of previous studies: flap-twist to feather coupling increases the frequency and reduces the damping, and flap-twist to stall coupling reduces the frequency and increases the damping of the flapwise mode. Flap-twist coupling has little influence on frequency and damping of the edgewise mode. Flap-twist to feather coupling reduces the blade root flapwise moment frequency response to mean wind speed variation which concurs with fatigue load reduction that have been observed for flap-twist to feather coupled blades. The frequency response of the tower bottom fore-aft moment is also reduced for flap-twist to feather coupled blades. The effect of flap-twist coupling on the classical flutter and divergence speeds could not be investigated as the first edgewise blade mode of the DTU 10 MW RWT becomes unstable before those speeds are reached.

*Acknowledgements.*  The present work is funded by the European Commission under the programme 'FP7-PEOPLE-2012-ITN Marie Curie Initial Training Networks' through the project 'MARE-WINT - new MAterials and REliability in offshore WINd Turbines technology', grant agreement no. 309395.





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
