# Peer review of "Modal Properties and Stability of Bend-Twist Coupled Wind Turbine Blades"

_Wind Energy Science, 2016_

## Referee Comment (RC1) · Anonymous Referee #1 · 7 Jan 2017

General comments: Overall a well-written and interesting paper which covers much ground on investigating the dynamics of bend-twist coupling on a large turbine (10 MW).

Specific comments:

1. Section 2- is any of this repeated in Stablein 2016a/b? If so, does it need to be repeated?

2. Sec 2.5.1 any explanation for the differences in the higher modes?

3. Sec 2.5.4 define gamma_y- is done later but should be here.

4. Sec 4.1 can you explain the sudden slope change in the tips for Fig 5/6

5. Sec 4.1.3 I am having difficulty understanding the phase angle relationships, the

work of the lift force, and the damping. Can this probably be described in more detail?

6. Sec 4.3 not clear how sensitivity to pitch is shown in Fig 12.

7. Sec 4.3 can you comment on the dip in Figure 18?

8. Sec 5 not clear how damping is shown in Fig 4.

Technical corrections:

1. Eq 3 and elsewhere- define length L- for the element or the entire blade. Eq 6 implies that that polynomial coefficients do not vary along L.

2. Fig 12/13/14/16 caption- should be amplitude (left column) phase (right column)

3. Sec 4.3 line 17- change camping to damping. No fun being dampened while camping.

4. Sec 5 "An observation that has also..." this sentence is not clear.

---

## Referee Comment (RC2) · Anonymous Referee #2 · 15 Mar 2017

In the paper, the authors study the impact of structural bend-twist coupling (flap-twist to feather, flap-twist to stall, edge-twist to feather, and edge-twist to stall) on the natural frequencies and damping characteristics of wind turbine rotor blades, and the aeroelastic stability of said rotor blades. The aeroelastic stability is also studied on turbine level in order to account for the dynamic interaction between the turbine components. The authors assume the bend-twist coupling to be triggered by the anisotropic nature of fiber composite material usually used for wind turbine rotor blades. For the description of the structural dynamics, a finite beam element formulation based on generalized degrees of freedom allowing for full coupling between the degrees of freedom is utilized. The aerodynamics is accounted for by means of unsteady blade element momentum theory. The models have been implemented in the HAWCStab2 code, which is a turbine simulation tool in the frequency domain, allowing for the calculation of the rotation

speed dependent frequency and damping characteristics. The structural model is thoroughly validated against several numerical examples from literature. For the studies, the DTU 10 MW reference turbine was chosen as a baseline, and the blades were modified to include the bend-twist coupling characteristics. The couple blades are compared to the baseline configuration.

The topic of aeroelastic stability and the proper calculation of frequency and damping characteristics is highly relevant for the wind energy science community, especially for very large rotor blades such as those of the DTU 10 MW reference wind turbine. Due to the very high impact of aeroelastic stability on the structural health of wind turbines, the topic is highly important and of broad international interest.

The paper presents results achieved by simulations with a commercially available tool which was extended by a finite beam element formulation that was previously published. The simulation results as such are novel, interesting, not obvious in all concerns, important, and highly relevant. It is not clear if the results are generally valid, or turbine specific, but that is a general issue in wind energy research.

The objectives are clearly formulated and addressed continuously throughout the paper. The paper is generally very well written, well structured, and clearly formulated to the point. The language is fluent, precise, and grammatically correct.

The scientific methods are valid, state of the art, and well chosen. They are clearly described and reproducible. The discussion of analysis results seems valid and detailed. The presented results form the basis for discussions at a later stage of the manuscript, and support the interpretations without exceptions. The discussion is relevant and backed up by own results and investigations done by other authors. The conclusions reached are accurate and base on the presented results. The authors also give proper credit to related and relevant work from other authors, and clearly indicate their own contributions and also construct the context to other works properly.

The title is chosen to the point. It reflects the content of the paper, is informative,

and gives the reader the chance to find the paper when searching for respective content. The abstract provides a concise and complete summary, and includes qualitative results (which makes sense in the context of shortness). Quantitative results are presented at a later stage of the manuscript.

Figures and tables are useful and all necessary. However, some of them could be positioned closer to where they are discussed in the text. Mathematical formulae, symbols, abbreviations, and units are mainly correctly defined and used according to the author guidelines. The number and quality of references is appropriate.

Some exceptions from the aforementioned comments are present in the paper. The following minor changes should be implemented in the manuscript:

- Section 2.5.4: The parameter $\gamma$y is introduced in section 3, but is utilized here already. Please introduce symbols where first used.

- Table 3: It would be nice not only to see the difference in numbers, but to know about the principle differences in beam element formulation without reviewing the entire references. Some basic remarks will be welcome.

- In order to save space, the authors should place figures 3 and 4 side by side, as well as figures 5 and 6.

- The authors should make sure that the figures are included in the text close to where referenced. Sometimes, the reader has to turn several pages, which is no fun at

- Figures 5 and 6, and partly Figures 12-14: The slope change at the tip looks erroneous (as it was hinged). Could the authors explain the reason?

To wrap up: The paper is definitely worth publishing. A proper implementation of the preceding comments will be appreciated. There is no need to re-review the manuscript.

---

## Author Comment (AC1) · 12 Apr 2017

Dear reviewer,

Thank you for your comments and constructive feedback. Please find our responses below:

1.) Section 2- is any of this repeated in Stablein 2016a/b? If so, does it need to be repeated?

- No, the previous publications are based on a different beam formulation.

2.) Sec 2.5.1 any explanation for the differences in the higher modes?

- The three beam elements are based on different formulations. How those formulations effect the eigenmodes of coupled beams has not been investigated.

3.) Sec 2.5.4 define gamma_y- is done later but should be here.

- We will consider this in the revision of the paper.

4.) Sec 4.1 can you explain the sudden slope change in the tips for Fig 5/6

- The slope change is probably related to the double curvature of the blade at the tip. We can observe the same behaviour using another code (HAWC2). We are currently investigating this and will address the issue in the revision of the article.

5.) Sec 4.1.3 I am having difficulty understanding the phase angle relationships, the work of the lift force, and the damping. Can this probably be described in more detail?

- The relationship of phase, lift and damping has been treated extensively in Stäblein 2016a. However, for better readability we will include a section on aerodynamic damping analysis of blade modes in the revision.

6.) Sec 4.3 not clear how sensitivity to pitch is shown in Fig 12.

- The correct reference is Fig. 9. This will be corrected in the revision.

7.) Sec 4.3 can you comment on the dip in Figure 18?

- The dip around 0.25 Hz is caused by antiresonance due to the interference with the tower fore-aft mode. We will add an explanation in the revision.

8.) Sec 5 not clear how damping is shown in Fig 4.

- The correct reference is (cf. Figure 9 and 4). This will be corrected in the revision.

Technical corrections:

1.) Eq 3 and elsewhere- define length L- for the element or the entire blade. Eq 6 implies that that polynomial coefficients do not vary along L.

- L is the length of the element. We will clarify this in the revision.

2.) Fig 12/13/14/16 caption- should be amplitude (left column) phase (right column)

**WESD**

- Yes. We will correct this in the revision.

3.) Sec 4.3 line 17- change camping to damping. No fun being dampened while camping.

- Yes. Will be corrected.

4.) Sec 5 "An observation that has also..." this sentence is not clear.

- We will change the sentence to: An effect of blade deflection on the edgewise mode shape has also been observed by Kallesøe and Hansen (2009).

---

## Author Comment (AC2) · 12 Apr 2017

Dear reviewer,

Thank you for your comments and constructive feedback. Please find our responses below:

1.) Section 2.5.4: The parameter $\gamma y$ is introduced in section 3, but is utilized here already. Please introduce symbols where first used.

- We will consider this in the revision of the paper.

2.) Table 3: It would be nice not only to see the difference in numbers, but to know about the principle differences in beam element formulation without reviewing the entire references. Some basic remarks will be welcome.

[Figure]

- We will add the following section: Hodges et al. (1991) uses finite beam elements based on a mixed variational formulation with cross-sectional properties obtained by a virtual work method by Giavotto et al. (1983). The formulation by Armanios and Badir (1995) is based on a variational asymptotic method and Hamilton's principle.

3.) In order to save space, the authors should place figures 3 and 4 side by side, as well as figures 5 and 6.

- We will consider this in the revision of the paper.

4.) The authors should make sure that the figures are included in the text close to where referenced. Sometimes, the reader has to turn several pages, which is no fun at

- We will move them closer to the text.

5.) Figures 5 and 6, and partly Figures 12-14: The slope change at the tip looks erroneous (as it was hinged). Could the authors explain the reason?

- The slope change is probably related to the double curvature of the blade at the tip. We can observe the same behaviour using another code (HAWC2). We are currently investigating this and will address the issue in the revision of the article.

---

## Author Response (AR1)

Dear Professor Peinke,

Please find attached our revised manuscript. We have made the following changes:

**Page 4, line 3:**
Element length introduced as suggested by reviewer #1.
**Page 6, lines 8 ff:**
Coupling coefficients introduced earlier in the manuscript as suggested by reviewer #1 and #2.
**Page 7, lines 1 ff:**
Short description of the different element formulations added as suggested by reviewer #2.
**Page 7, line 12:**
Sentence has been reformulated to make use of the coupling coefficient introduced earlier.
**Page 8, Table 1 caption:**
'FEM model' has been replaced by 'shell model' to be more specific.
**Page 9, line 3:**
Tense corrected.
**Page 9, lines 8 ff:**
A section on aerodynamic damping analysis of blade modes has been added to allow an easier interpretation of the results as suggested by reviewer #1.
**Page 11, lines 3 ff:**
The definition of the coupling coefficients has been removed as they are now introduced earlier in the manuscript.
**Page 11, line 26:**
Grammatical correction.
**Page 12, lines 9 ff:**
We have investigated the kink at the blade tip pointed out by both reviewers and added an explanation to the text.
**Page 12, line 14:**
The statement '(and the gradients around the linearisation point)' is not correct and has been removed.
**Page 18, Figure 12; Page 19, Figure 13; Page 20, Figure 14 and Page 22, Figure 16:**
Caption corrected as suggested by reviewer #1.
**Page 21, Figure 15:**
Caption changed to be in line with other figures.
**Page 21, Line 4:**
Reference corrected as pointed out by reviewer #1.
**Page 21, Line 8:**
'whirling' added for better understanding.
**Page 21, Line 16:**
Typo corrected as pointed out by reviewer #1.
**Page 22, Line 7:**
Explanation of the frequency dip in the frequency response added as requested by reviewer #1.
**Page 23, Figure 17 caption:**
Typo corrected.
**Page 23, Line 3:**
Reference corrected as pointed out by reviewer #1.
**Page 25 Line 2:**
Sentence has been rephrased for better understanding as suggested by reviewer #1.

Kind regards,

Alexander Stäblein

[revised manuscript text omitted]